# Functional and Multi-Omics Effects of an Optimized CRISPR-Mediated FURIN Depletion in U937 Monocytes

**DOI:** 10.3390/cells13070588

**Published:** 2024-03-28

**Authors:** Ruiming Chua, Lijin Wang, Roshni Singaraja, Sujoy Ghosh

**Affiliations:** 1Program in Cardiovascular and Metabolic Disorders, Duke-NUS Medical School, Singapore 169857, Singapore; ruiming_chua@duke-nus.edu.sg; 2Centre for Computational Biology, Duke-NUS Medical School, Singapore 169857, Singapore; lijin.wang@duke-nus.edu.sg; 3Yong Loo Lin School of Medicine, National University of Singapore, Singapore 119077, Singapore; mdcrrs@nus.edu.sg; 4Laboratory of Computational Biology, Pennington Biomedical Research Center, Baton Rouge, LA 70808, USA

**Keywords:** myeloid cells, FURIN protease, gene editing, functional assays, genomics, cytokine secretion

## Abstract

The pro-protein convertase FURIN (PCSK3) is implicated in a wide range of normal and pathological biological processes such as infectious diseases, cancer and cardiovascular diseases. Previously, we performed a systemic inhibition of FURIN in a mouse model of atherosclerosis and demonstrated significant plaque reduction and alterations in macrophage function. To understand the cellular mechanisms affected by FURIN inhibition in myeloid cells, we optimized a CRISPR-mediated gene deletion protocol for successfully deriving hemizygous (HZ) and nullizygous (NZ) *FURIN* knockout clones in U937 monocytic cells using lipotransfection-based procedures and a dual guide RNA delivery strategy. We observed differences in monocyte and macrophage functions involving phagocytosis, lipid accumulation, cell migration, inflammatory gene expression, cytokine release patterns, secreted proteomics (cytokines) and whole-genome transcriptomics between wild-type, HZ and NZ FURIN clones. These studies provide a mechanistic basis on the possible roles of myeloid cell FURIN in cardiovascular disorders.

## 1. Introduction

The pro-protein convertase, FURIN (PCSK3) belongs to a family of proteins that affect a wide variety of biological functions through proteolytic cleavage and the activation of target proteins [1,2]. Through genetic association analyses, we previously identified *FURIN* as a gene candidate associated with coronary artery disease in humans [3]. This genetic finding was subsequently validated in two independent mouse models of atherosclerosis, where the systemic inhibition of FURIN was found to provide atheroprotection as well as reduce restenosis [4]. An associated finding was that FURIN inhibition reduced the number of macrophages in the atherosclerotic plaques of the coronary artery, suggesting that FURIN inhibition may play a specific role in myeloid cell function. Notably, FURIN expression is increased in several cell types in human atherosclerotic lesions, including macrophages [4,5]. Based on these observations, we wanted to examine the effects of *FURIN* gene editing on key functions in cultured monocytes/macrophages.

Based on the innate immune response mechanisms in bacteria, clustered regularly interspersed short palindromic repeats/CRISPR-associated protein 9 (CRISPR/Cas9) gene-editing technologies have evolved as a powerful and popular methodology for genome editing both in vivo and in vitro. When two or more sgRNAs are co-expressed with a single Cas9 protein, different genomic loci can be targeted simultaneously, thereby greatly improving the efficiency of gene editing, including gene deletions and the generation of hemizygous (HZ, containing a wildtype and a mutant allele) and nullizygous (NZ, both copies of genes deleted) clones [6,7,8]. These advantages have greatly popularized the CRISPR/Cas9 systems as indispensable tools in the field of genome editing and have shown great promise even for gene-targeted therapy [9].

THP-1 and U937 are two of the most commonly studied pro-monocytic human leukemia cell lines that are capable of differentiating into macrophages or dendritic cells in vitro [10]. However, the phagocytic properties of monocytes often lead to the degradation of exogenously introduced DNA, accompanied by cell death [11]. The three inter-related problems of low transfection efficiency, low cell viability and low success of stable transfections make monocytes/macrophages refractory to DNA transfections in general, including CRISPR-mediated genome editing [11,12]. Some advanced methods exist, but they generally require special equipment that is often expensive and proprietary, and therefore not available in all research settings. Additionally, they are also often associated with significant cell toxicity. There remains a need for optimizing simpler techniques to enable successful genome editing in normally refractory cell types such as monocytes.

In this manuscript, we report on an optimized protocol for CRISPR editing in U937 monocytes using standard lipid-based transfection reagents, resulting in hemizygous and nullizygous clones for *FURIN* and observed important differences in several targeted cellular phenotypes as a function of *FURIN* gene dosage. Additionally, we carried out whole-genome transcriptomic and targeted proteomic characterizations of HZ and NZ *FURIN* clones compared to wild-type cells (WT), to characterize the effects of FURIN on myeloid cell function on a broader scale.

## 2. Materials and Methods

### 2.1. Cells and Reagents

U937 cells (catalog no. CRL-1593.2,) and THP-1 monocytic cells (catalog no. TIB-202) were purchased from ATCC (Manassas, VA, USA). Plasmids for CRISPR-related cell transfection were a gift from Dr. Shang Li, Duke-NUS. gRNAs and PCR primers were purchased from IDT (Singapore). RPMI media, Opti-mem, L-glutamine and lipofectamine were purchased from ThermoFisher (Singapore). Fetal bovine serum was purchased from Research Instrument (Singapore). The 96-well plates for cell culture were obtained from Practical Mediscience (Singapore), and 24- and 6-well plates were from Scimed (Asia) (Singapore). DAPI solution was purchased from BD (Franklin Lakes, NJ, USA). PCR reactions were performed on a CFX96 (Bio-Rad, Hercules, CA, USA) using PCR mastermix from Promega (Madison, WI, USA). Western blotting was performed using the Semi-Dry Rapid Blot Transfer system from Bio-Rad.

### 2.2. Plasmid Preparation

Two single-guide RNAs (sgRNA) targeting the 5′ and 3′ ends of the *FURIN* genomic region (chr15:91411885-91426687, hg19) were designed using an online software (https://zlab.bio/guide-design-resources from crispr.mit.edu (accessed on 30 August 2018), currently unavailable). The sgRNA sequences were as follows (gRNA1-Forward: 5′—CCAAGGAGACGGGCGCTCCA—3′; gRNA1-Reverse: 5′—TGGAGCGCCCGTCTCCTTGG—3′; gRNA2-Forward: 5′—GCCGGGGCAAGCTGCCCTAA—3′; gRNA2-Reverse: 5′—TTAGGGCAGCTTGCCCCGGC—3′). The plasmid construct is shown in Appendix A, and the expected cleavage sites in the *FURIN* gene are shown in Appendix A. The engineering of a single vector co-expressing spCas9-T2A-GFP and the two sgRNAs was performed as described previously [13,14]. Briefly, the plasmids pX330A-1x2 Cas9-2A-GFP and pX330S-2 were digested with BbsI restriction enzyme. The sgRNA1 was inserted into the pX330A-1x2 Cas9-2A-GFP plasmid, while the sgRNA2 was inserted into the pX330S-2 plasmid. The U6 cassette (U6-sgRNA2-gRNA scaffold) from pX330A-1x2 was then inserted downstream of the U6 cassette (U6-sgRNA1-gRNA scaffold) pX330A-1x2 Cas9-2A-GFP vector using a BsaI restriction enzyme following the golden gate assembly procedure [13]. This results in the GFP being cloned to the C-terminal of CRISPR plasmid (PX330A Cas9-2A), such that GFP is expressed only when the full-length plasmid had been successfully transcribed from its N- to C-terminal. Cells that express GFP were therefore expected to also contain the CRISPR Cas 9 gene product, as well as the two gRNAs transcribed (Appendix A).

### 2.3. Cell Culture, Transfection and Cell Proliferation Assays

THP1 and U937 monocytes were cultured in RPMI 1640 media containing L-glutamine with 10% FBS and 1% pen-strep. Cells were passaged every 3 days on average prior to transfection. Both U937 and THP1 cells were transfected for 24 h without changing the media followed by screening for GFP expression via florescence microscopy (Appendix A). A Bio-Rad cell counter TC20 was used to perform all cell counting. Nuclear counterstaining was performed via DAPI. GFP-positive cells were sorted into single cells into a 96-well plate based on DAPI-negative (preferentially stained dead cells) and GFP-positive staining, as determined by FACS. In all transfection assays, Hek293 cells were used as a positive control. Cell proliferation was monitored through a Cell Titre Glo reagent (Promega), as per the manufacturer’s protocol, on 1000 cells/100 μL in 96-well plates, and measurements taken at 24 h intervals after seeding.

### 2.4. PCR Screening for CRISPR

The PCR screening of the transfected clones was performed using a set of internal and external primers designed around the *FURIN* gene. The primer sequences were as follows: (Internal Forward: 5′ 5′—GACGGCTACACCAACAGTATC—3′; Internal Reverse: 5′—CACAGAGTGCCTTCTACCTAAC—3′; External Forward: 5′—AGTCTTCATCCTGCTTCTTCT—3′; External Reverse: 5′—GCCTGGATGGGACCATTATT—3′). For internal primers (expected amplicon size 993 bp), the standard PCR protocol was used. For PCR involving external primers (expected amplicon size ~7.3kb) the GoTag Long PCR kit from Promega (cat: M4021) was used, and PCR was performed according to the manufacturer’s instructions.

### 2.5. Western Blot

Proteins were separated in 10.5% polyacrylamide gels containing Sodium Dodecyl Sulfate (SDS) and transferred to PVDF membranes (Bio-Rad) using the semi-dry blotting system (Bio-Rad) and the Transblot Turbo 5X Transfer buffer (all from Bio-Rad), following the manufacturer’s protocol. After the transfer, the membrane was blocked with 5% milk (Bio-Rad) for 1 h at room temperature. The membranes were incubated with a rabbit monoclonal anti-FURIN antibody (1:5000 dilution, Cell Signaling, Danvers, MA, USA) overnight. The blots were washed three times with Tris-buffered saline with 0.1% Tween-20 (TBST) and then incubated with a goat anti-mouse polyclonal antibody (1:10,000 dilution) conjugated with horseradish peroxidase (HRP) for 1 h at room temperature (BD). Subsequently, they were visualized using enhanced chemiluminescence (Pierce).

### 2.6. Lipid Uptake Assay

Lipid uptake by the macrophages was performed using Dil-Ox LDL (L34358) obtained from Life Technology, following the manufacturer’s protocol. U937 cells were differentiated into macrophages using 100 nM phorbol 12-myristate 13-acetate (PMA) for 72 h. Subsequently, they were exposed to fluorescently labeled oxidized lipids (Dil-Ox LDL) for 4–8 h. The extent of cellular lipid uptake per cell (intensity density) was captured through fluorescent imaging using a Leica microscope (Wetzlar, Germany) and then quantified using ImageJ software [15] by comparing the lipid-associated red fluorescence to the fluorescence from nuclear staining with Hoechst 33,342 stain.

### 2.7. Phagocytosis Assay

Phagocytosis assays were performed using a Phrodo green *E. coli* conjugate obtained from Life Technologies (Carlsbad, CA, USA). Prior to the assay, U937 cells were differentiated into macrophages using 100 nM PMA for 72 h. Phagocytosis assays were conducted as per the manufacturer’s instructions. The extent of phagocytosis was quantified through the ImageJ software 1.51 by comparing green fluorescence from engulfed *E. coli* to the fluorescence from nuclear staining with Hoechst 33,342 stain.

### 2.8. Migration Assay

In the transwell cell migration assay, we employed Transwell 6.5 mm inserts with a 24-well plate configuration, containing 8.0 μm polycarbonate membranes (Sigma, St. Louis, MO, USA). The lower chamber was loaded with 600 μL of media containing 10% FBS and 10 nM CXCL12 (SDF-1α) as a chemoattractant sourced from Sigma, with an incubation duration of 8 h. The upper chamber was seeded with 150,000 U937 monocytes suspended in 100 μL of media lacking FBS. The number of migrated cells was quantified using the Cell Counting Kit-8 from Dojindo (Kumamoto, Japan).

### 2.9. Inflammatory Gene Expression via Quantitative PCR Assays

Total RNAs were extracted from the WT, HZ and NZ *FURIN* clones of U937 cells using TRIzol (Life Technologies). Single-strand cDNA was prepared via reverse-transcription using random hexamers using the IScript cDNA synthesis kit (Bio-Rad). Gene expression was measured via quantitative RT-PCR (CFX96) using the ssoAdvUniversal Sybrgreen master mix (Bio-Rad). Both the signals and relative gene expression were normalized to corresponding cyclophilin gene expression controls. The fold change in gene expression was analyzed using the 2^(−ΔΔCt) method [16]. The primer sequences for the qPCR for all genes are listed in Appendix A.

### 2.10. RNA Sequencing Analysis

Total RNA was prepared from triplicate cell cultures of WT, HZ and NZ *FURIN* clones of U937 cells, as described above. RNA sequencing was performed at Novogene (Singapore) on the Illumina Novaseq 6000 platform with the 150-base, paired-end sequencing strategy.

The quality of the RNA-seq reads was ascertained via FASTQC (http://www.bioinformatics.babraham.ac.uk/projects/fastqc (accessed on 10 December 2023)). The median sequencing depth was 52 million reads per sample, with a median per-base quality >30 for all samples. Adapter trimming was performed via BBDuk (https://jgi.doe.gov/data-and-tools/software-tools/bbtools/bb-tools-user-guide/bbduk-guide/ (accessed on 10 December 2023)). Sequencing reads were mapped to the human reference genome (GRCh38) via STAR (2.7.7.a) [17], with an average mapping rate of 74.5%. Raw count matrices of RNA sequencing data were obtained through the featureCounts [18] package in R and further processed for the gene quantification and identification of differentially expressed genes using the limma package [19]. Gene counts were log2 transformed and normalized for sequencing depth using the trimmed means of M-values (TMM) method [20]. The TMM-normalized data were used to identify potential sample outliers via principal component analysis through the princomp package in R. Genes with at least one count per million (CPM) reads in three or more samples were retained for further analysis, resulting in 13,582 genes. The mean–variance relationship of the gene-wise standard deviation to average log CPM gene signal was assessed using ‘voom’ [21] to generate precision weights for each observation, which were then used to generate empirical Bayes moderated t-statistic estimates for the identification of differentially expressed genes. To adjust for multiple tests, adjusted *p*-values were calculated using the false discovery rate (FDR) [22]. Genes with absolute fold changes ≥2-fold and adjusted *p*-values < 0.01 were considered differentially expressed. The full RNA sequencing data were deposited to the Gene Expression Omnibus (GEO, https://www.ncbi.nlm.nih.gov/geo/info/ (accessed on 10 December 2023)) with series accession number GSE248467.

### 2.11. Pathway Enrichment Analysis

Pathway enrichment analysis on differentially expressed genes was conducted via gene set enrichment analysis (GSEA) [23], using pathway databases from the KEGG and Gene Ontology Biological Process (GOBP), obtained from MSigDB [24]. GSEA was run in pre-ranked mode with genes ranked by their log fold change between comparisons. The analysis was restricted to pathways containing between 15 and 250 genes, and the enrichment statistic was computed using the ‘classic’ method. Pathways with an adjusted *p*-value ≤ 0.25 were considered to be significantly regulated.

### 2.12. Self-Organizing Map (SOM) Analysis

SOM analysis was performed on a subset of differentially expressed genes, including genes with a minimum adjusted *p*-value < 0.01, a maximum fold change > 2-fold and maximum average group expression (logCPM) > 2 across HZ vs. WT and NZ vs. WT comparisons. A total of 1515 genes met these conditions. The SOM analysis was carried out using the som package in R (https://cran.r-project.org/web/packages/som/index.html (accessed on 10 December 2023)) by setting the neighborhood parameter to ‘Gaussian’, along with a rectangular topology and linear initiation. A subset of the SOM gene clusters was further analyzed for biological pathway enrichment through the Bioplanet database [25] in the Enrichr enrichment analysis tool [26]. We also queried two additional databases in Enrichr (‘ENCODE_and_ChEA_Consensus_TFs_from_ChIP-X’, and ‘TF_perturbations_followed_by_expression’) to identify putative transcription factors of which their regulation through the cellular FURIN status could give rise to the gene expression patterns observed across the different SOM clusters.

### 2.13. Targeted Proteomic Analysis

Quantitative sandwich-based antibody arrays (RayBio^®^ Human Cytokine Array C3) were used to quantify the levels of 42 cytokines secreted from the WT, HZ and NZ FURIN clones of U937 cells, before and after stimulation with LPS (5 ug/mL, overnight stimulation). Briefly, 1 mL of media from each experiment were added into each array blot and incubated with capture antibodies overnight at 4 °C. After washing, the arrays were incubated with a biotin-conjugated anti-cytokine antibody mix overnight at 4 °C. HRP-conjugated streptavidin was added to bind with biotin from the detection antibodies for 2 h at room temperature, and the chemiluminescent signal was detected using a Universal Hood III (Bio-Rad). After 40 s of exposure, signal values for each spot were captured and quantified using ImageJ. For each blot, background counts were subtracted from the raw counts of each spot. Background-corrected counts were further normalized using a scaling factor derived by equalizing the average intensity of the positive control spots from each blot. The background-subtracted, scaled spot intensities were used for linear regression modeling to identify significant changes in protein levels as a function of LPS treatment and FURIN genotype, along with any interaction effects between them.

### 2.14. Data Analysis

For all experiments, with the exception of the quantitative PCR and RNA sequencing analyses, data were summarized through the calculation of the means and standard deviations for each assay and presented either as bar charts or boxplots. Statistical significance for between-group comparisons was assessed using heteroscedastic *t*-tests with the threshold for statistical relevance set at *p* ≤ 0.05. The analysis methods for the qPCR and RNA sequencing data have been described in their respective sections above.

## 3. Results

### 3.1. Transfection Efficiency and Post-Transfection Viability of THP-1 and U937 Monocytes

To compare transfection efficiencies, both THP-1 and U937 cells were transfected with 1.5 μL of lipofectamine and 1 μg of plasmid for 24 h (Figure 1a). Although the overall transfection efficiency was very low, U937 cells demonstrated a better transfection efficiency (0.71%) compared to THP1 cells (0.54%), as determined by the number of GFP-positive cells post transfection (Figure 1b,c). The FACS-estimated mean GFP expression signal was also higher in U937 cells (7289.44 units) compared to THP1 cells (1237.56 units), suggesting a greater expression of GFP (and therefore, the co-transcribed gRNAs) in U937 cells (Figure 1d). At 24 h post transfection, cell debris was observed in the transfected THP1, but not in the U937 cells. Cell viability assessment via Trypan blue staining showed a significant reduction in THP1 cell growth (Figure 1e) at 24 and 48 h post transfection, compared to the U937 cells.

### 3.2. Transfection Optimization

To optimize the transfection efficiency for U937, several transfection conditions were tested. Optimization experiments were conducted with 120,000 cells seeded into each well of 24-well plates in a final volume of 500 μL. The amount of lipofectamine 3000 was varied from 0.75 μL to 1.5 μL, whereas the transfectant plasmid was used at 1 μg, 3 μg or 5 μg per transfection. The amount of ‘lipo-complex’ that was added into each well was also tested at 50 μL and 75 μL volumes. All experiments were conducted in triplicate, and experiments were further duplicated on two different days. Of all the conditions tested, the largest transfection efficiency (0.71%) was observed with 1.5 μL lipofectamine 3000 with 1 μg plasmid and 50 μL of lipo-complex (Figure 2). Increasing the plasmid amount to 3 μg or 5 μg always led to lower transfection, at all levels of lipofectamine or lipo-complex tested. We used a non-myeloid, human embryonic kidney cell line (HEK-293T) as a positive control for transfection and observed it to have a transfection efficiency of 54% and cell viability of 86% at 24 h post transfection. Although the transfection efficiency in U937 cells was much lower than that for the easily transfectable HEK-293T cells, the observations were reproducible and led to the successful identification of HZ and NZ knockouts of the FURIN gene in U937 cells.

### 3.3. PCR Screening of CRISPR Candidate Clones

GFP-positive clones were seeded into 96-well plates at one cell/well and allowed to grow for 2 weeks before screening for gene deletion via PCR. RNA was prepared from candidate clones and subjected to PCR analysis using external and internal primer pairs as described earlier. A schematic of the primer locations with respect to the FURIN gene is shown in Figure 3a, and the PCR results from 18 candidate clones are shown in Figure 3b. Of these, one clone was identified as a NZ knockout of FURIN (lane 10 from the left, lane 1 being molecular weight markers), and five clones were putative HZ clones (lanes 11,12,15,16, and 18), with one copy of the gene deleted. Further confirmation of the HZ and NZ clones was obtained through a direct DNA sequencing of the PCR fragments (Appendix A).

### 3.4. Validation of NZ and HZ Clones via qPCR and Protein Blotting

We quantified the FURIN mRNA expression in WT, HZ and NZ U937 clones via quantitative PCR. Interestingly, the level of FURIN message was similar between the WT and HZ clones, whereas no message could be detected in the NZ clone (Figure 3c). FURIN protein levels were assessed via Western blotting on cell lysates from the WT, HZ and NZ clones using a goat anti-human anti-FURIN antibody (Figure 3d). The FURIN protein expression was observed in the WT and HZ clones (lanes 1, 2), but no signal was detected in the NZ clone (lane 3). We compared the growth rates of the WT, HZ and NZ FURIN clones and found that while the proliferations of WT and HZ clones were comparable, the total absence of FURIN in the NZ clones was associated with a reduced rate of growth. Thus, over 4 days, the growth rate of the NZ clone was less than 50% of the rates observed for the WT and HZ clones, suggesting that a complete lack of FURIN is detrimental to cell growth (Figure 3e).

### 3.5. Loss of FURIN Does Not Modulate Macrophage Phagocytosis

A comparative analysis of the phagocytic capacity was performed on the WT, HZ or NZ FURIN clones (differentiated into macrophages). The results are shown visually in Figure 4a (green fluorescence from phagocytosed *E. coli*, blue fluorescence from Hoechst 33,342 nuclear staining) and further quantified in Figure 4b. No significant differences were observed in the abilities of these clones to phagocytose fluorescently labeled *E. coli*, suggesting FURIN activity to be dispensable for this function (*p* > 0.05 for all comparisons to WT).

### 3.6. Loss of FURIN Reduces Macrophage LDL Accumulation

A comparison of the abilities of the WT, HZ and NZ FURIN clones to accumulate lipid was performed by first differentiating the clones into macrophages and then exposing them to fluorescently labeled oxidized LDL particles for 4–8 h. The results suggest a reduced, but not statistically significant, lipid accumulation capacity in the HZ clone compared to that of the WT (*p* > 0.05). However, a significantly reduced lipid accumulation was observed in the NZ clone (*p* < 0.05), suggesting a potential role of FURIN in regulating lipid accumulation in macrophages (Figure 5a,b).

### 3.7. Loss of FURIN Reduces Monocyte Trans-Migration

We quantified the number of U937 monocytes (WT, HZ or NZ for FURIN) undergoing transmembrane migration to a chemoattractant (CXCL12) through a transwell migration assay. Beginning with the same number of cells (150,000), the HZ FURIN clone showed a statistically non-significant reduction compared to the WT clone at all timepoints tested. However, there was a significant impairment of cell migration in the NZ FURIN clone when compared to WT U937 cells, beginning from 4 h of migration (Figure 6). This suggests a clear role of FURIN in the monocyte migration capacity that is mostly preserved even with one functional copy of the FURIN gene.

### 3.8. Loss of FURIN Shows Variable Effects on Inflammatory Gene Expression

WT, HZ and NZ U937 monocyte clones were stimulated with LPS for up to 6 h, followed by a quantitative PCR analysis of selected the inflammatory gene expression in cell lysates. The changes in gene expression fold changes varied widely, ranging from a less than 4-fold change for CD68 to over a 100-fold change observed for IL-6 (6 h values), although for several genes, noticeable changes in gene expression were observed only at the 6 h timepoint (Figure 7). The pattern of gene expression also varied between genes and could be classified into three main classes. Genes such as CD68, COX1 and AMAC1 showed a similar time-dependent increase in gene expression in all samples (compared to expression levels in WT samples at baseline), independent of the gene-editing status of FURIN. Genes such as IL-6 and IL-8 displayed significantly higher changes in expression in WT samples compared to both types of FURIN-edited clones. Interestingly, for TGFB and LPL, the HZ clone showed maximal changes at 6 h followed by the NZ and WT clones.

### 3.9. RNA Sequencing Analysis

#### 3.9.1. Loss of FURIN Impacts Transcriptomic Landscape

The RNA sequencing analysis identified significant differences in the transcriptome expressions across the three clone types. The principal component analysis showed a clear separation between the WT, HZ and NZ FURIN monocyte clones along the first principal component, with the HZ clones clustering between the WT and NZ clones (Appendix A). Nearly 49% of the variation in gene expression was captured in the first principal component, suggesting the FURIN copy number to be a major driver in the transcriptomic landscape. A further separation of the HZ clones with respect to the WT and NZ clones was observed in the second principal component, accounting for nearly 25% of the residual gene expression variance. A total of 968 and 2019 genes were differentially expressed when comparing WT samples to HZ and NZ FURIN clones, respectively, based on a cutoff of an at least two-fold up- or down-regulation and an adjusted *p*-value of 0.01, suggesting a progressively increasing effect on gene transcription due to the loss of one or both copies of FURIN. An overlap comparison of the differentially expressed genes showed 674 genes to be regulated in common in both the HZ and NZ clones compared to WT, whereas 294 and 1345 genes were uniquely regulated in HZ and NZ compared to WT, respectively (Figure 8a). The extent of the differential gene expression is depicted through volcano plots for WT vs. HZ FURIN clones compared to WT cells (Figure 8b,c). The per-sample expression of the top 50 most differentially expressed genes (selection based on adjusted *p*-value) are further depicted through heatmaps (Figure 8d,e) and show that while about an equal number of genes were up- and down-regulated between WT and HZ clones, a very large majority (47 out of 50) of the top differentially expressed genes were down-regulated in NZ clones compared to WT cells (additional details of the transcriptome analysis are provided in Appendix A).

#### 3.9.2. Loss of FURIN Affects Biological Pathways

In order to identify relevant biological mechanisms enriched among the differentially expressed genes, we performed a gene set enrichment analysis (GSEA) on the KEGG and GOBP pathways (Appendix A). For the HZ vs. WT comparison, five KEGG pathways and one GOBP pathway were significant at an adjusted *p*-value < 0.2, whereas for NZ vs. WT, four KEGG and two GOBP pathways were significant at the same statistical threshold (Figure 8f). Of these, the KEGG pathway termed ‘Complement and Coagulation Cascades’ was significantly downregulated in both the HZ and NZ clones compared to the WT cells. We highlighted the genes contributing to the enrichment of this pathway (based on their fold changes observed in the HZ vs. WT and NZ vs. WT samples) on a KEGG map of the pathway (Figure 8g,h). From the figure, it is clear that genes in both the ‘intrinsic’ and ‘alternative’ arms of the complement cascade were differentially regulated in both comparisons, suggesting a broad effect of FURIN on this pathway.

We further enquired the pathway enrichment results to determine if the cellular effects observed earlier in the WT, HZ and NZ clones (cell proliferation, phagocytosis, lipid accumulation and trans-migration) were also reflected in the respective cellular transcriptomes (Appendix A). We observed increases in the proliferative pathways in both the HZ and NZ clones compared to WT (nominal *p*-value < 0.002), but the NZ clone also demonstrated significant increases in apoptosis, which helps explain the observed reduction in the growth rate in NZ. A pathway related to the regulation of lipid storage was down-regulated in both the HZ and NZ clones although the effect was statistically relevant only in the HZ clone. We did not observe any significant enrichment in phagocytic pathways in either HZ or NZ, which is in agreement with the results obtained from cellular phagocytic studies. Finally, both HZ and NZ clones demonstrated statistically relevant reductions in chemotactic pathways, consistent with the observed differences in these clones for trans-migration studies. Thus, the observed cellular phenotypes were generally supported by the attendant transcriptomic changes in the HZ and NZ clones, compared to cells with control FURIN levels.

#### 3.9.3. Loss of FURIN Leads to Varying Gene Expression Patterns

In order to identify the patterns of gene expression as a function of the copy number of FURIN, we performed a self-organizing map (SOM) analysis on a subset of genes (minimum adjusted *p*-value ≤ 0.01, maximum absolute fold change ≥ 2.0 and maximum average logCPM ≥ 2 across the three groups) and identified eight distinct patterns of gene expression clusters (Figure 9a). Individual gene-cluster associations are provided in Appendix A. We performed a pathway enrichment analysis using Enrichr on the SOM clusters and identified the enrichment of several pathways (adjusted *p*-value ≤ 0.1) among the different patterns (Figure 9b–f). Pathways such as the ‘Complement and Coagulation Cascades’ and ‘extracellular matrix organization’ were enriched in clusters 2–3, where the gene expression was progressively decreased with a reduction in or loss of FURIN, whereas pathways related to TGF-1 beta signaling, IL-1 signaling and chromosome maintenance were enriched for genes that had an increased expression upon a reduction in FURIN (clusters 6–8). The full list of pathways impacted by the loss of FURIN is available in Appendix A.

### 3.10. Loss of FURIN Variably Modulates Cytokine Secretion

We investigated changes in selected cytokine secretion from U937 cells as a function of the FURIN copy number, either in basal or LPS-stimulated conditions. Figure 10 shows five key categories of changes in secretion levels for selected cytokines under basal and LPS stimulation in each of the WT, HZ and NZ clones (the full results for all cytokines are available as Appendix A). For each cytokine, the statistical significance of the secretion differences was estimated using linear regression models. Compared to NZ, the top five cytokines showing a statistically significant difference of secretion in HZ under baseline and LPS-stimulated conditions included ANG (reduced in both baseline and LPS), TNF alpha, MDC and EGF (increased in both baseline and LPS) and GMCSF (increased in baseline and reduced under LPS). Similarly, the top five cytokines showing significant differences in secretion between WT and NZ cells included angiogenin, SDF1, IL6 (reduced in both baseline and LPS), TARC and SCF (reduced in baseline and increased in LPS). The top five cytokines affected by LPS stimulation independent of the FURIN genotype included TARC, TGFB, Leptin (reduced in LPS), GMCSF (reduced in WT and HZ, increased in NZ), and SCF (increased in WT, reduced in HZ and NZ). In addition to these main effects, interaction effects were also observed for some cytokines. For example, the top five statistically significant HZ–LPS interactions were observed for GMCSF, GCSF, PDGFBB, ENA78 and TGFB, whereas the top five interactions for WT–LPS were observed for GMCSF, TARC, SCF, IL2 and ENA78. Overall, we observed heterogeneity in cytokine secretion in response to the FURIN copy number. The full results from the linear modeling are provided in Appendix A. We further compared the cytokines on the array to a predicted list of FURIN substrates (PMID 23335997) and identified overlap for only three cytokines (PDGFB, TGFB and IGF1). For both TGFB and IGF1, a loss of FURIN (HZ or NZ) resulted in increased cytokine secretion under basal conditions that was suppressed by LPS stimulation. For the other cytokines, the observed effects are presumed to be secondary, not related to their proteolytic cleavage through FURIN (although the possibility of them being unknown FURIN substrates cannot be ruled out).

## 4. Discussion

Through proteolytic cleavage and the activation of diverse protein substrates in the secretory pathway, intracellular FURIN acts as a central regulator of several biological functions. Cell-surface expressed FURIN also cleaves a diverse array of pathogenic substrates including bacterial toxins and viral fusion peptides [27,28], thereby providing protection from infections. Conversely, FURIN activity can also aid in viral infection, the most notable recent example being that of coronavirus SARS-CoV-2 [29]. Together, these make FURIN an attractive target for therapeutic intervention in multiple disorders [1,30,31]. Previously, we identified FURIN as a coronary artery disease-associated gene candidate based on a genetic association analysis of the CardiOgram cohort [3,32]. We further demonstrated that the systemic inhibition of FURIN in mouse models of atherogenesis resulted in a lower atherosclerotic lesion area and a reduction in severe lesions, whereas the exogenous introduction of FURIN protein led to increased intimal plaque thickness [4]. Additionally, we observed reduced monocyte migration and reduced inflammatory and cytokine gene expression in macrophages upon FURIN inhibition. Although our studies, and others, clearly point to a pro-atherogenic role of FURIN [33,34,35,36], some studies have reported contrasting findings [37,38]. The differences probably arise from differences in the model systems used and may possibly also reflect changing roles for FURIN dependent on the stage and extent of atherosclerosis. These differences prompted us to investigate the role of FURIN in a monocyte/macrophage cell line in greater detail. Although phagocytic cells are refractory to transfections, we developed an optimized lipid-based transfection protocol to successfully perform CRISPR-driven gene editing in U937 cells, resulting in cells containing either heterozygous or homozygous knockouts of the *FURIN* gene.

An examination of several cellular functions in U937 cells with a wild-type (WT), hemizygous (HZ) or nullizygous (NZ) FURIN status revealed several interesting observations. Compared to WT cells, the cell-growth rate was similar in HZ but reduced in NZ cells. Cell viability was also similar between WT and HZ cells but reduced in NZ cells, suggesting a role of FURIN in macrophage cell proliferation and survival. However, the reduction in or loss of FURIN did not affect the phagocytic abilities of the cells, suggesting that the inhibition of FURIN spares this innate immunity function of macrophages related to the ingestion and degradation of bacteria, dead cells, debris, tumor cells and foreign materials [33]. We carried out additional experiments to determine if the FURIN status impacts the differentiation of monocytes to macrophages, based on the expressions of monocyte (CCR2) and macrophage (CD68, ADGRE1) marker genes during the time course of differentiation. Our results suggest no significant differences in the monocyte-to-macrophage differentiation process between the WT, HZ and NZ clones (Appendix A).

One key function of monocytes/macrophages related to atherosclerosis involves the scavenging of oxidized lipids and transformation into foam cells, followed by the production of proinflammatory mediators [39]. We therefore tested the effects of *FURIN* gene dosage on oxidized lipid accumulation through U937 cells and observed that the accumulation of scavenged lipids was reduced in HZ and NZ cells (although the reduction in HZ was not statistically significant). Another important early event in atherosclerosis is the increased recruitment of monocyte-derived cells into the subendothelial space where they subsequently differentiate into macrophages and contribute to the dynamic progression of atherosclerotic plaques [40]. We tested the abilities of the WT, HZ and NZ U937 clones for trans-well migration and observed consistently reduced migrations for both HZ and NZ clones at all timepoints tested (although the reduction in HZ was not statistically significant). The selected inflammatory gene expression analysis via qPCR of U937 clones identified a heterogeneity of gene regulation in response to LPS stimulation, with genes such as IL6 and IL8 showing marked reductions in transcription in response to the loss of one of both copies of FURIN, whereas IL23, LPL and TGFB displayed increased transcriptions in both the HZ and NZ clones compared to WT.

A broader global transcriptome profiling of the WT, HZ and NZ clones identified clear differences in gene transcription as a function of the FURIN copy number, with progressively increasing effects on gene transcription caused by the loss of one or both copies of the FURIN gene. A pathway enrichment analysis identified biological mechanisms related to complement and coagulation being significantly downregulated in both the HZ and NZ clones compared to WT, which is suggestive of alterations in key macrophage functions involving innate immunity and hemostasis [41,42]. Additionally, an analysis of the gene expression patterns based on self-organizing maps identified pathways related to TGF-1 beta signaling, IL-1 signaling and chromosome maintenance to be enriched among genes with higher expressions in HZ and NZ compared to WT clones. Finally, a focused analysis of the secreted proteome revealed differing patterns of cytokine release from the WT, HZ and NZ U937 clones under basal and LPS-stimulated conditions. Notably, FURIN depletion resulted in an increase in the secretion of IGF-1, a key factor associated with inflammation resolution in macrophages [43,44,45] and directly implicated in reducing atherosclerotic burden in mice [36,37]. Another finding of potential importance is the up-regulation of ANG (Angiogenin) in FURIN-depleted cells, given that endothelial ANG has been recently identified as an anti-atherogenic factor [46].

In summary, we had previously shown that FURIN inhibition decreased atherosclerotic lesions in two independent mouse models. We showed here that some pathways by which FURIN inhibition may confer atheroprotection is by reducing monocyte migration, macrophage proliferation and foam cell formation, and by reducing specific atherogenic inflammatory cytokine production. We further show global-scale transcriptomic and transcriptionally dictated pathway alterations as a consequence to FURIN inactivation. Finally, a proteomic analysis revealed alterations in the secretion of cytokines, some of which are directly implicated in atherogenic processes. These results provide further insights into the functions of FURIN in myeloid cells and suggest that monocyte/macrophage-specific targeted FURIN inhibition may be a useful therapeutic strategy for atherosclerosis and related disorders.

## Figures and Tables

**Figure 1 cells-13-00588-f001:**
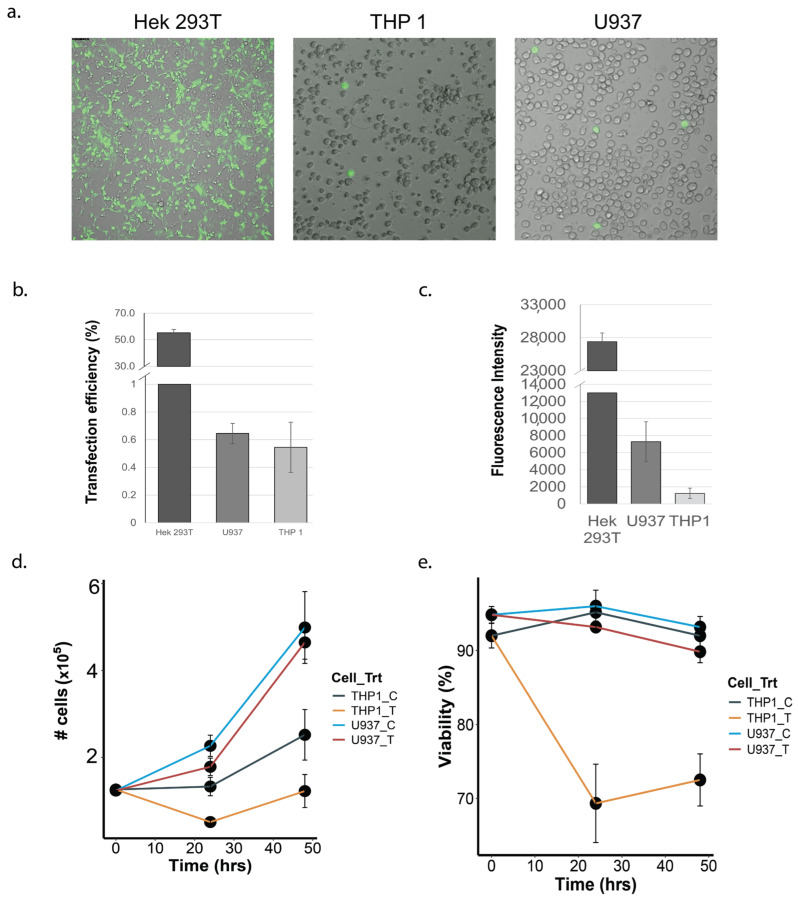
Comparison of transfection efficiency in monocytic and non-monocytic cell-lines. (**a**) GFP fluorescence indicating the extent of the transfection of the GFP-containing plasmid vector in Hek293T, THP1 and U937 cells. The Hek293T cells were used as a positive transfection control and displayed robust plasmid transfection, whereas transfection was detectable but much lower in the monocytic cell lines. (**b**) Quantification of transfection efficiencies (percent of cells transfected) in Hek293T, THP1 and U937 cells. (**c**) Comparison of fluorescence signals (copies of GFP per cell) between Hek293T, THP1 and U937 cells, showing a greater GFP expression in U937 cells compared to THP1 cells. (**d**) Effect of transfection on cell proliferation. Untransfected (U937-C, THP1-C) and transfected (U937-T, THP-T) cells were monitored for cell proliferation over a period of 48 h post transfection. (**e**) Effect of transfection on viability of U937 and THP1 cells. Untransfected (U937-C, THP1-C) and transfected (U937-T, THP-T) cells were followed for 48 h post transfection, and the number of viable cells were quantified via Trypan blue staining. For both (**d**,**e**), the color-coding refers to distinct cell line–treatment combinations.

**Figure 2 cells-13-00588-f002:**
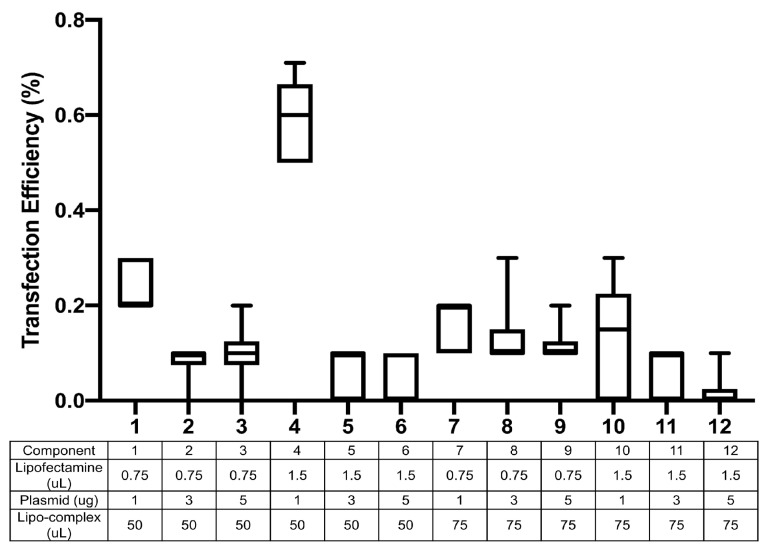
Optimization of transfection in U937 cells. Different combinations of the transfection agent (lipofectamine), transfection plasmid and the lipo-complex were tested, and the efficiency of transfection was monitored through the GFP expression in U937 cells. The *x*-axis indicates each of the 12 conditions tested as noted in the table below the plot. Transfection efficiency (% cells transfected) is quantified on the *y*-axis. All experiments were conducted in triplicate and further duplicated on two different days.

**Figure 3 cells-13-00588-f003:**
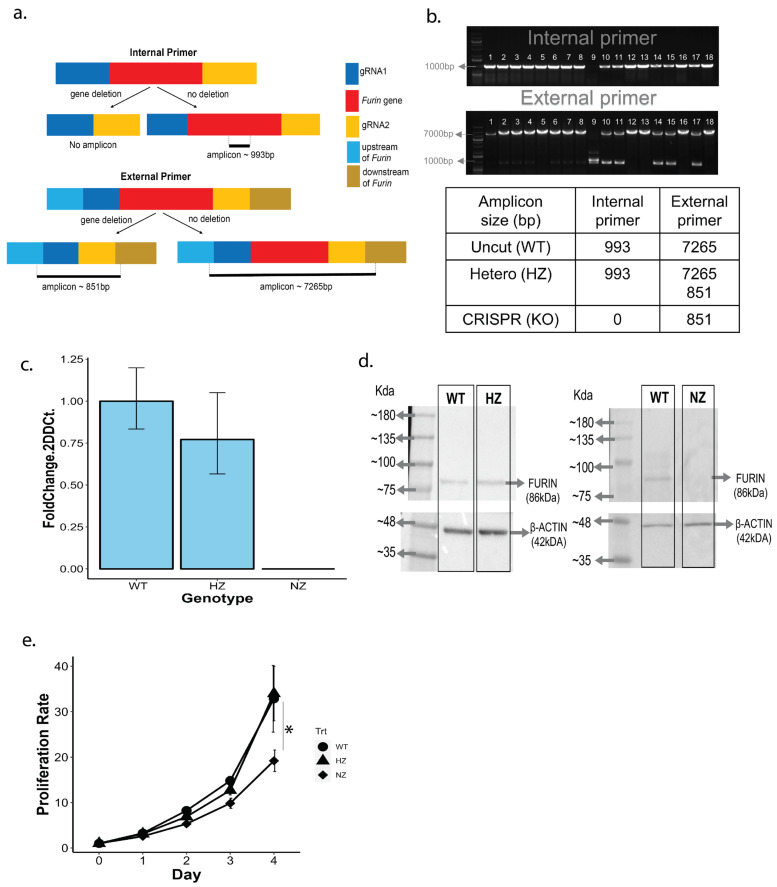
Identification of CRISPR-mediated U937 clones hemizygous (HZ) and nullizygous (NZ) for FURIN, and detection of FURIN transcript and protein. (**a**) Schematic of the internal primer (internal to the FURIN coding region) and external primers (external to and flanking the FURIN gene) used to test for the presence of CRISPR-mediated FURIN gene deletion via PCR. The various segments around the FURIN gene are color-coded and explained through the legend on the right. For both internal and external PCR primers, the expected amplicon length in the presence or absence of FURIN gene deletion are indicated through the black bars. (**b**) Results from PCR analysis of a selection of 18 CRISPR-edited U937 clones. The top panel depicts PCR amplicons generated by the internal primer, and the bottom panel depicts amplicons generated by the external primer. Molecular weight markers are shown in lane 1 of each electropherogram. Lane 9 in both images correspond to a homozygous FURIN deletion (NZ), whereas lanes 10, 11, 14, 15 and 17 refer to potential heterozygotes with one copy of the FURIN gene intact (HZ). Other lanes refer to clones with a wild-type FURIN status. The expected amplicon sizes for wild-type (WT), hemizygous (HZ) and nullizygous (NZ) FURIN statuses are shown in the table below. (**c**) Quantification of FURIN messenger RNA in WT, HZ and NZ cell clones via qPCR. Results are depicted as fold changes with respect to the clone containing wild-type FURIN, and were calculated using the delta-delta Ct method. (**d**) Western blot analysis of FURIN protein expression in WT, HZ and NZ clones. For each blot, the molecular weight markers are indicated on the leftmost lane. The left panel shows the FURIN expression in the WT and HZ clones, along with that of beta-ACTIN as a loading control. The right panel quantifies the FURIN expression in the WT and NZ clones along with beta-ACTIN controls (no FURIN expression was observed in the NZ sample). (**e**) Quantification of cell proliferation in WT (circle), HZ (triangle) and NZ (diamond) U937 clones over 96 h post seeding. Statistical significance of differences in cell proliferation was ascertained by conducting *t*-tests at each time-point (*, *p* < 0.05).

**Figure 4 cells-13-00588-f004:**
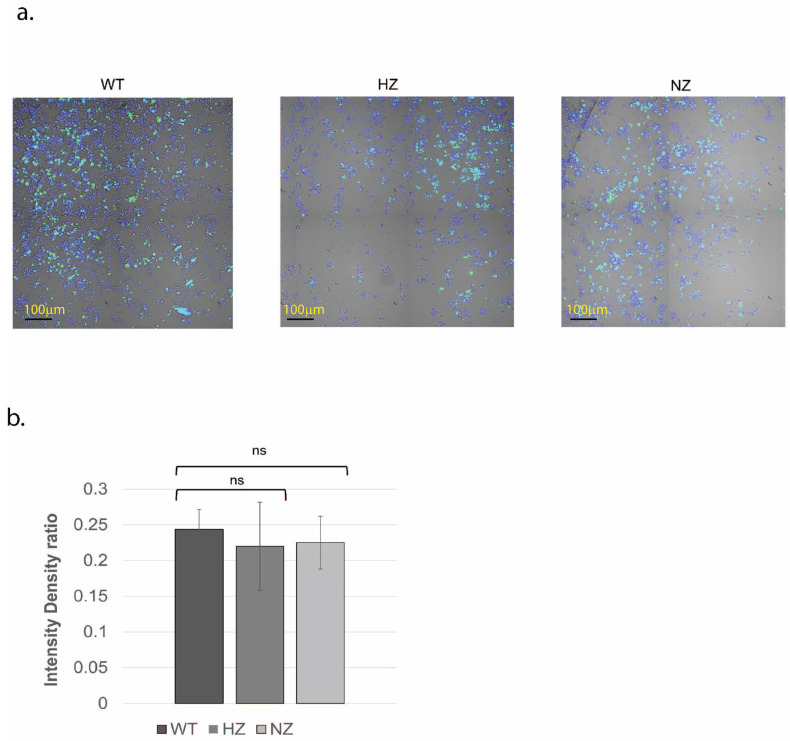
Effects of FURIN gene status on phagocytic activity of U937 cells. (**a**) WT, HZ and NZ U937 clones were differentiated into macrophages and tested for engulfment of pHrodo green E. coli bioparticles conjugate. Nuclear staining was performed via Hoechst 33342. (**b**) Quantification of phagocytosis through intensity density ratio plot (pHrodo fluorescence/nuclear fluorescence). Experiments were conducted in triplicate. Statistical significance of differences in phagocytic activity was ascertained using *t*-tests (ns, *p* > 0.05).

**Figure 5 cells-13-00588-f005:**
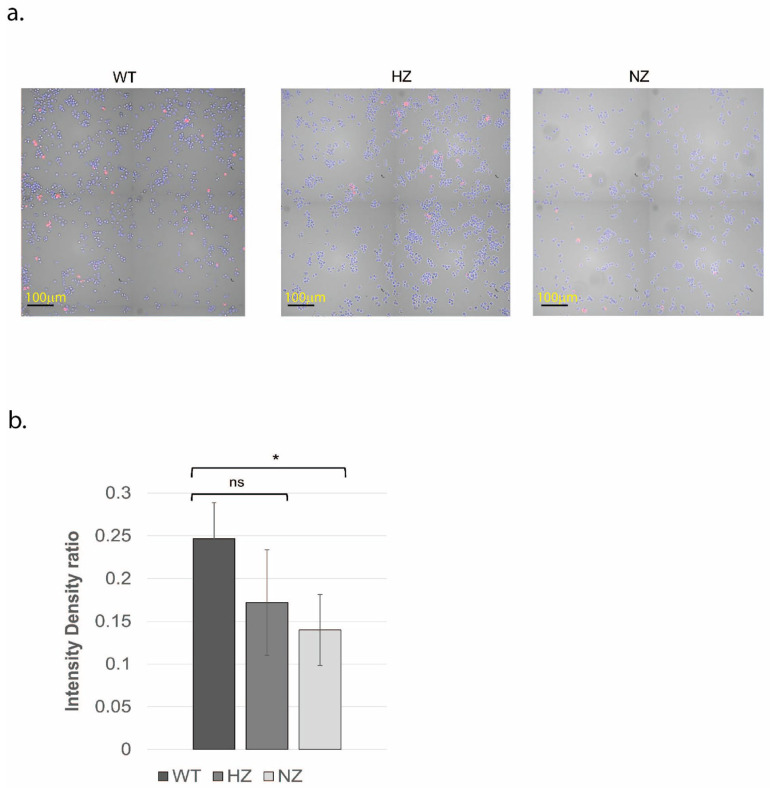
Uptake of oxidized lipid as a function of FURIN gene status. (**a**) WT, HZ and NZ U937 clones were differentiated into macrophages and exposed to fluorescently labeled oxidized LDL particles for 4–8 h. Nuclear counterstaining was performed via Hoeschst 33,342. (**b**) Quantification of oxidized lipid uptake through intensity density ratio plot (pHrodo fluorescence/nuclear fluorescence). Experiments were conducted in triplicate. Statistical significance of differences in oxidized lipid uptake was ascertained using *t*-tests (*, *p* < 0.05; ns, *p* > 0.05).

**Figure 6 cells-13-00588-f006:**
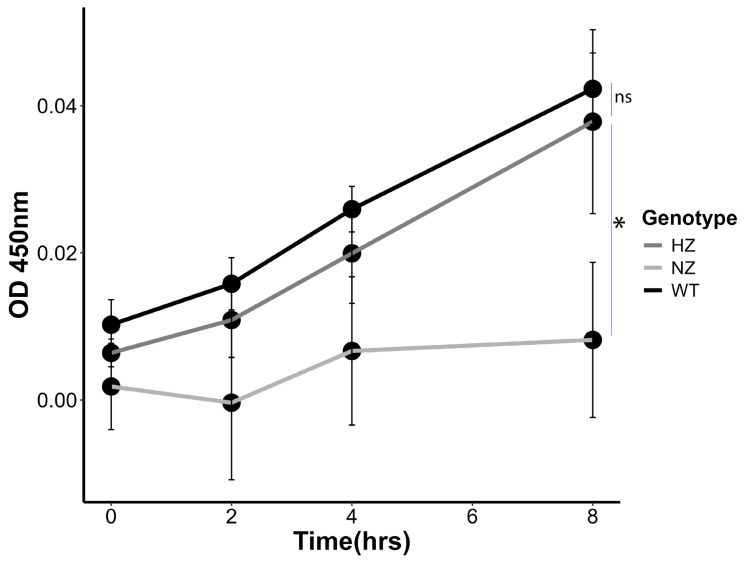
Effect of FURIN gene status on chemoattractant-induced migration. WT (black line), HZ (deep gray line) and NZ (light gray line) U937 monocyte clones were seeded in a transwell chamber and subjected to CXCL12-induced migration over 8 h. The number of cells migrating to the lower chamber of the transwell was quantified using Cell Counting Kit 8. Experiments were conducted in triplicate. Statistical significance of differences in trans-migration was ascertained using *t*-tests (*, *p* < 0.05; ns, *p* > 0.05).

**Figure 7 cells-13-00588-f007:**
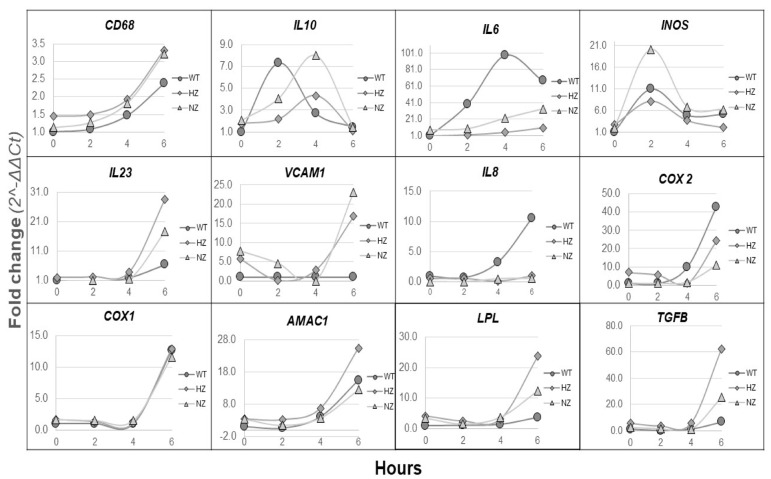
Effects of the FURIN gene status on the inflammation-related gene expression. WT (gray circle), HZ (gray diamond) and NZ (gray triangle) U937 monocyte clones were assayed for the expressions of selected inflammation-related genes following LPS-stimulation, via qPCR. All results are reported as the fold change in gene expression with respect to the basal 0 h expression. Experiments were conducted in triplicate, and fold changes in gene expressions were calculated using the delta-delta Ct method. HZ, heterozygous; NZ, homozygous knockout (nullizygous).

**Figure 8 cells-13-00588-f008:**
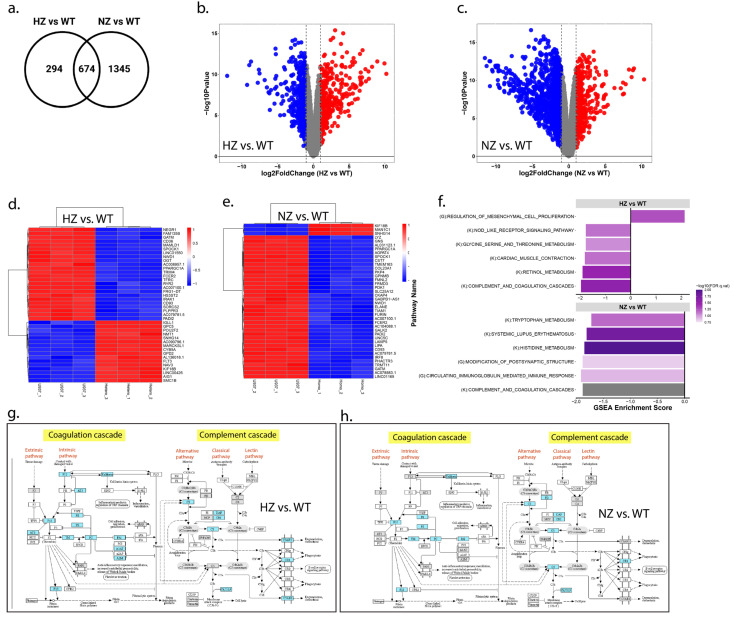
Effect of FURIN the gene status on the whole-genome transcriptome expression in U937 cells. (**a**) Unique and overlapping differentially expressed genes (absolute fold-change ≥ 2, adj.*p*-value ≤ 0.01) in the HZ and NZ clones compared to in WT. (**b**,**c**) Volcano plots of log2 fold-change (x-axis) vs. -log10 *p*-value in HZ vs. WT and NZ vs. WT comparisons, respectively. Genes with >=2-fold upregulation are colored red, genes with <=2-fold downregulation are colored blue, and other genes are shown in gray. (**d**,**e**) Heatmaps of the top 40 most differentially expressed genes (sorted by adjusted *p*-value) in HZ vs. WT and NZ vs. WT comparisons, respectively. (**f**) Bar chart depicting the top up- and down-regulated KEGG and Gene Ontology Biological Process pathways identified via gene set enrichment analysis (adjusted *p*-value ≤ 0.2) of HZ vs. WT and NZ vs. WT. The pathway enrichment scores are plotted on the x-axis, and pathway names are listed on the y-axis. Each pathway name is preceded by a (K) or (G) to indicate the source of the pathway as KEGG or GOBP, respectively. Bars are color-coded according to the −log10 of the adjusted *p*-value of the pathway enrichment. (**g**,**h**) KEGG pathway maps for the ‘Complement and Coagulation Cascades’ pathway in HZ vs. WT and NZ vs. WT, respectively, with the genes contributing to pathway enrichment highlighted in blue.

**Figure 9 cells-13-00588-f009:**
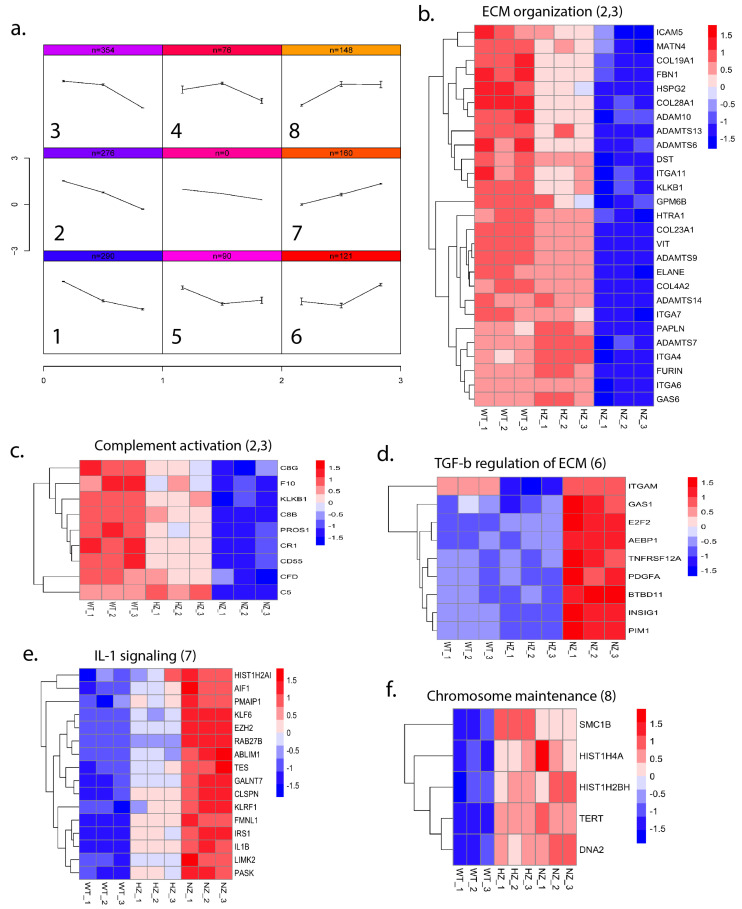
Patterns of gene expression as a function of the FURIN gene status. (**a**) Gene expression profiles across WT, HZ and NZ U937 cells were clustered using self-organizing maps (SOMs), depicting eight distinct patterns. The number of genes for each pattern are listed at the top of each subplot. (**b**–**f**) Pathway enrichment analysis of gene sets in selected clusters. Each heatmap plots the expression of cluster-specific genes (rows) across replicate samples (columns). Expression values are row-normalized and color-coded from low (blue) to high (red) expression. The pathway name is indicated at the top of each heatmap, and the SOM cluster number is indicated in parentheses.

**Figure 10 cells-13-00588-f010:**
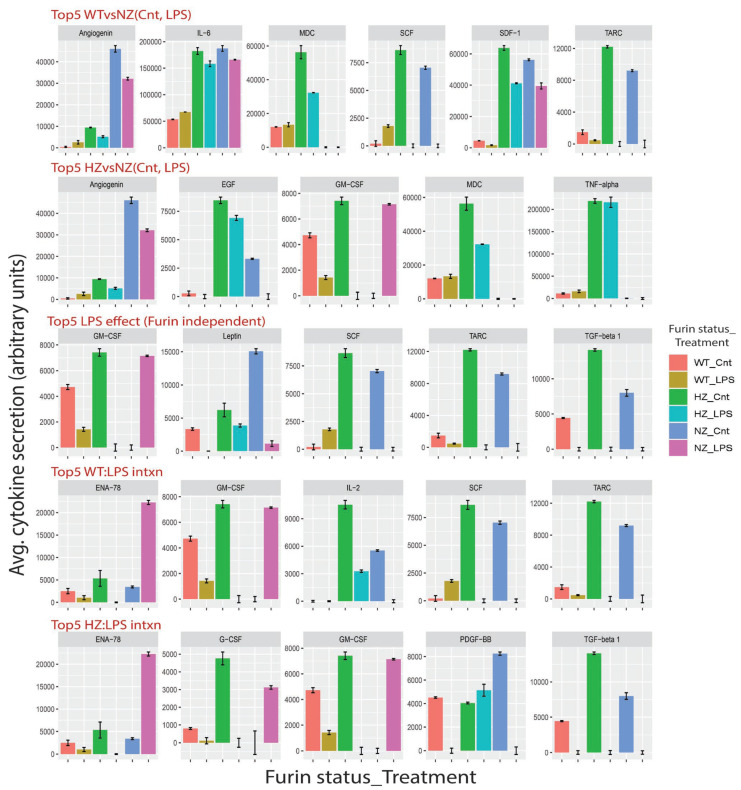
Proteomics analysis of the effects of the FURIN gene status on cytokine secretion under basal and LPS-stimulated conditions. Cytokine secretion was assessed on antibody arrays and quantified after normalization via background subtraction and scaling. The plot is divided into five panels representing key cytokine secretion patterns observed in response to LPS stimulation and FURIN gene status (WT, HZ, NZ): Top5 WTvsNZ (Cnt, LPS), top five cytokines showing significant changes in secretion between WT and NZ samples under both basal and LPS stimulation; Top5 HZvsNZ (Cnt, LPS), cytokines showing significant secretion differences between HZ and NZ clones under both basal and LPS stimulation; Top5 LPS effect (Furin independent), top five cytokines showing significant secretion effects upon LPS stimulation, regardless of the FURIN gene status; Top5 WT:LPS intxn, top five cytokines showing significant interaction effects between WT and LPS stimulation; Top5 HZ:LPS intxn, top five cytokines with significant interaction effects between HZ and LPS stimulation. Cytokine names are indicated at the top of each subplot. Bars are color-coded by a combined identifier containing the FURIN gene status and LPS stimulation status. The *x*-axis lists the combined identifiers, and the *y*-axis indicates the normalized cytokine secretion levels (results are averaged over duplicates).

## Data Availability

The data presented in this study are publicly available on Github (https://github.com/sg3451/FURIN_U937_study/tree/main accessed on 10 December 2023). Gene expression data was submitted to GEO with accession number GSE248467.

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
