# Peer review of "Functional and Multi-Omics Effects of an Optimized CRISPR-Mediated FURIN Depletion in U937 Monocytes"

_cells, 2024, doi:10.3390/cells13070588_

Round 1

Reviewer 1 Report

Comments and Suggestions for Authors

Chua R and others performed CRISPR-Cas9 knockout of FURIN in human monocyte cell line U937. The authors successfully established the HZ and NZ FURIN edited cell lines and performed a series of functional assays and transcriptome analysis to characterize the phenotypes associated with the gene dose of FURIN in U937. Overall the conclusions were supported by the evidence presented in the manuscript. I only have some minor suggestions:

  1. When performing macrophage-related functional assays, no data demonstrates the cells were fully differentiated into macrophages. Does FURIN editing affect macrophage differentiation/maturation/polarization? Gene expression or flow cytometry analysis using different macrophage stage markers can comprehensively characterize macrophage differentiation.

  2. RNAseq and functional assays seem not to support each other. Are there any genes related to phagocytosis, proliferation, or migration altered in the edited cells? 

  3. It is great that the authors performed proteomics analysis on the secreted cytokines with/without LPS stimulation. Yet, the Figure 10 is hard to read and follow. It would be better to select some of the cytokines and highlight some of the changes that match the description in the manuscript.

Reviewer 2 Report

Comments and Suggestions for Authors

The study optimized lipid-based transfection protocol to successfully perform CRISPR-driven gene editing in refractory phagocytic cells for FURIN protein which is previously identified by authors as gene candidate associated with coronary artery disease in human. Authors examined the impaction of FURIN deletion in monocytes/macrophages cultures using wide range of molecular, biochemical, and immunological techniques. The Methodology part is well written with good order. The results part is clear however, some figures need more resolution. The discussion and conclusion are well supported by results. 

Minor comments:

1-   I think that using (pro-monocytic) in the title is not appropriate here, while using monocytes cell line is more approachable.

2-   In Figure 8 (f-j): I can understand the authors need to show all data in this figure, but it is really hard to see the details here. I suggest more highly resolute images or putting them as supplementary figs. 
